# Intelligent Simulation of Water Temperature Stratification in the Reservoir

**DOI:** 10.3390/ijerph192013588

**Published:** 2022-10-20

**Authors:** Yuan Yao, Zhenghua Gu, Yun Li, Hao Ding, Tinghui Wang

**Affiliations:** 1College of Civil Engineering and Architecture, Zhejiang University, Hangzhou 310058, China; 2Nanjing Hydraulic Research Institute, Nanjing 210029, China; 3State Key Laboratory of Hydrology-Water Resources and Hydraulic Engineering, Nanjing 210029, China

**Keywords:** water temperature stratification, artificial neural networks, reservoir, intelligent simulation, Tankeng hydropower station

## Abstract

In order to fully make use of limited water resources, humans have built many water conservancy projects. The projects produce many economic benefits, but they also change the natural environment. For example, the phenomenon of water temperature stratification often occurs in deep reservoirs. Thus, effective ways are needed to predict the water temperature stratification in a reservoir to control its discharge water temperature. Empirical formula methods have low computational accuracy if few factors are considered. Mathematical model methods rely on large amounts of accurate hydrological data and cost long calculation times. The purpose of the research was to simulate water temperature stratification in a reservoir by constructing an intelligent simulation model (ISM-RWTS) with five inputs and one output, determined on the basis of artificial neural networks (ANN). A 3D numerical model (3DNM) was also constructed to provide training samples for the ISM-RWTS and be used to test its simulation effect. The ISM-RWTS was applied to the Tankeng Reservoir, located in the Zhejiang province of China, and performed well, with an average error of 0.72 °C. Additionally, the Intelligent Computation Model of Reservoir Water Temperature Stratification (ICM-RWTS) was also discussed in this paper. The results indicated that the intelligent method was a powerful tool to estimate the water temperature stratification in a deep reservoir. Finally, it was concluded that the advantages of the intelligent method lay in its simplicity of use, its lower demand for hydrological data, its well generalized performance, and its flexibility for considering different input and output parameters.

## 1. Introduction

Large-scale water conservancy projects, while generating huge benefits for flood control, irrigation, and power generation, can cause serious environmental and ecological problems [1]. Of these, thermal stratification is an important issue. Large-scale Reservoirs and dams can alter the natural thermal regimes in rivers [2,3], as well as contribute water pollution and dissolved oxygen concentrations into rivers and lakes, which are directly related to their water temperatures [4,5]. During large-scale water conservancy project planning and designing, water environment changes from thermal stratification, one of the most important factors, must be considered [6]. Water temperature stratification in reservoirs can result in some significant impacts on the water environment and aquatic ecosystem in a reservoir region and downstream river [7], which hamper the normal lives of local residents [8]. For example, they may lower the temperature of the downstream water resources used for agricultural irrigation [9]. Furthermore, aquatic organisms are sensitive to water temperature changes. Suitable water temperature is important for some fish species in downstream rivers, but the changed water temperature may affect their survival, growth and demographic characteristics [10]. The damage caused by discharged water with low temperatures from deep reservoirs is often called the ‘cold water disaster’ [11].

So far, the computational methods of reservoir water temperature stratification can be divided into two kinds: the empirical formula method and the numerical method [12]. The empirical formula method can only provide 1D profile distributions, and has low computational accuracy for the few factors are considered. The numerical methods include 1D, 2D and 3D models. WRE [13], MIT, and ‘Lake Temperature 1′ [14,15] are representative 1D water temperature models. In 1975, Edinger et al. [16] developed the LARM as the earliest 2D water temperature model. Then CE-QUAL-W2 was developed based on the LARM, which is a general 2D water temperature model [17]. During the end of the 20th century, commercial software based on three dimensional CFD bloomed rapidly, such as MIKE3, FLUENT, and DEFT3D. Compared to 1D models with rough results for predicting water temperature stratification, 2D or 3D numerical models are used more at present [12,18], although they are limited by hydrology data, topography data, and computation resources. In addition, researchers have introduced other methods to this field. Sahoo et al. [19] forecasted stream water temperature using regression analysis, artificial neural networks, and chaotic non-linear dynamic models. Diao et al. [20] predicted reservoir water temperature through the Lattice Boltzmann method (LBM). Buccola et al. [3] used streamflow response to precipitation (PRMS) to simulate river temperatures. Jackson et al. [21] investigated river temperatures using a spatio-temporal statistic model.

Since the 1980s, a mass of intelligent computing methods have been widely applied to simulation, prediction, optimization, and other scientific fields, such as artificial neural networks (ANN), genetic algorithms (GA), and fuzzy logic [22]. Therein, ANN have the abilities of information processing, self-learning, and reasoning, and show the characteristics of fault-tolerance, non-linearity, non-locality, non-convexity, and more [23]. These incomparable advantages over traditional methods make ANN very suitable for water science problems which are hard to establish effectively through formal models. In this study, the Intelligent Simulation Model of Reservoir Water Temperature Stratification (ISM-RWTS) was originally proposed on the basis of its ANN, and was verified by the case of the Tankeng Reservoir, locating in the Zhejiang province of China.

The paper was organized as follows: In Section 2, the study area was determined and the 3D numerical model (3DNM) of the Tankeng reservoir was established, which offered samples for the ISM-RWTS and was compared with the ISM-RWTS. In Section 3, the process of constructing the ISM-RWTS based on ANN was presented. In Section 4, the evaluation indicators of various given models were listed. In Section 5, the ISM-RWTS was applied to the Tankeng Reservoir and its performance was evaluated by comparing the results with the 3DNM and the measured results. Moreover, the Intelligent Computation Model of Reservoir Water Temperature Stratification (ICM-RWTS) was also proposed for isothermal prediction. Finally, a set of conclusions (Section 6) closed the paper.

## 2. Materials

### 2.1. Investigation Area

The Tankeng Reservoir (E120°02′, N28°06′), also known as Qianxia Lake, is located in the middle reach of the Xiaoxi Tributary of the Oujiang River in Qingtian County, Zhejiang Province, China. The watershed area above the dam is about 3330 km^2^, accounting for 93% of the total area. The length of the investigated river segment is about 87 km, and its average slope is 2.53‰. The watershed is long and narrow with a length of 105 km and an average width of 31.7 km. The Tankeng Reservoir’s average volume of runoff is 3.75 billion m^3^, its total storage capacity is 4.19 billion m^3^, and its normal water level is 160 m. The geographic location and water surface of the Tankeng Reservoir are shown in Figure 1. In terms of the effect of reservoir backwater, the hydrological condition was required to be close to that of the natural river while selecting the inlet of modeling domain. According to Lu’s calculation results [24] of the one-dimensional water surface curve of the Tankeng Reservoir, the modeling domain was finally determined as shown in Figure 1. The section 52 km away from the dam and the dam were selected as the inlet and outlet, respectively, of the investigation area, which were used in simulating the 3DNM.

### 2.2. 3D Numerical Model

The 3DNM of the Tankeng Reservoir was built using the CFD tool MIKE 3, which was developed by the Danish Hydraulic Institute (DHI). It is based on 3D incompressible Reynolds-averaged Navier–Stokes equations and assumptions of the Boussinesq and hydrostatic pressure [25]. In the 3DNM, developers also considered the influence of turbulent flow on the numerical solution and the variation of water density in the computational domain. The water density is closely related to the transport equations of salt and temperature. The model adopted the Smagorinsky turbulent diffusion module in the horizontal direction, and the k-ε turbulent diffusion module in the vertical direction.

The establishment of the 3DNM required a large number of grids to truly reflect the actual situation of the calculation area, but too many grids can greatly increase the calculating time. When we simulated the 3DNM for the first time, it cost about 700 h to finish this calculation in a four-core, 3.4 GHz computer. Due to the limitation in computer resources and massive simulating time, the research area needed to be smoothed on the edges. The smoothing greatly saved the simulation time and had nearly no influence on the calculation accuracy. After that, each simulation time was controlled within 300 h. The effects before and after smoothing are shown in Figure 2(a), Figure 2(b), respectively.

In the horizontal direction, the triangular unstructured grids were adopted. In the vertical direction, there are two kinds of meshing methods [26]: Sigma and Sigma-z. The Sigma method can accurately give the depth of the simulated area and provides consistent resolution at the riverbed, but over rapidly changing terrain, the Sigma coordinates create unreal flow which can have a negative effect on the horizontal pressure gradient calculations, convection, and mixing item. To accurately simulate the vertical flow, the Sigma-z method was finally adopted. As is shown in Figure 3, the model had 2423 grid nodes and 3630 grid cells. Two layers with the same thickness of 10 m were set as the Sigma grid for the depth of 20 m, while the other layers below 20 m were set as the z grid with 10 m interval.

The time step was set to 30 s. The calculation period ranged from January 15 to December 31 in 2016. The air temperature was set according to the local air temperature series in 2016. The solar radiation was set to the local radiation data. Relative humidity was set to the local annual average humidity of 76%. The wind speed was set to the local average speed of 1.3 m/s in 2016. Due to the limited hydrological data, we assumed that the effects of rainfall and evaporation offset each other. Initially, the initial water temperature was stationary with a uniform temperature of 12 °C, which referred to the measured water temperature of the reservoir on January 15. When the simulation started, the water flowed into the reservoir at a rate of 119 m³/s, which referred to the average flow rate in 2016, and was set as the temperature stratified inflow according to the measured data. The outlet drained at the same flow rate to maintain a constant water surface level. Based on the measured water temperature at 1 km away from the dam in the Tankeng Reservoir, the model was calibrated to determine the roughness height of the riverbed as 0.05 m, the horizontal diffusion coefficient, $D_{h}$, as 1 m^2^/s, and the vertical diffusion coefficient, $D_{v}$, as 0.05 m^2^/s.

## 3. Methodology

### 3.1. Topological Structure of ISM-RWTS

To construct the topological structure of the ISM-RWTS, it is important to limit the selection of parameters to those that reflect the most relevant characteristics affecting water temperature stratification. These temperature gradients result from diverse factors, such as local climate, topography of the reservoir, inflow, and outflow [20]. Figure 4 shows the main processes of heat exchange in reservoirs. The wind speed affects the surface heat exchange. The higher the wind speed, the higher the surface heat dissipation efficiency [27]. Air temperature and humidity influence the conduction heat loss and evaporation heat loss at the surface. Solar shortwave radiation and atmospheric longwave radiation penetrate deep into the reservoir, and a small proportion of energy is also reflected back into the atmosphere. At the bottom of the reservoir, the water and river bed also exchange heat at all times [28]. Inside the reservoir, the mixing of hot and cold water takes place on account of the disturbance of wind or the decrease in surface temperature. Moreover, inflow can bring huge amounts of heat or negative heat, while outflow can take away lots of heat or negative heat [29].

If there is no significant change in the operation rules of the reservoir, the variation rule of the water temperature stratification in the reservoir will remain stable during the period. In a year, the water temperature stratification during different seasons is greatly affected by the climate. Because tremendous changes in water temperature stratification in the reservoir often occur from May to September, to reflect the influence of climatic effects, including wind, solar radiation, and humidity, a total of five intelligent models were built for May, June, July, August and September in this study. The topography of the reservoir was described by some characteristic parameters, such as maximum depth and reservoir area. The outflow rate was considered equal to the inflow rate in order to reduce the number of inputs. With these factors in mind, a total of five parameters were finally selected as the ISM-RWTS’s inputs, i.e., reservoir depth (DR), reservoir area (AR), reservoir capacity (CR), water inflow (WI), and water depth (WD). Water temperature (WT) was selected as the ISM-RWTS’s output. Here, WD was defined as the depth from the water surface. In consideration of the ANN based on Error Back Propagation Algorithms (BPNN), which are mature and widely applied [30], we adopted it as the ISM-RWTS’s intelligent algorithm to identify the non-linear mapping relationship between the input and output. The ISM-RWTS’s topological structure is shown in Figure 5 and written as:(1)$$\mathrm{WT}=\mathrm{BPNN}\left(\mathrm{DR},\mathrm{AR},\mathrm{CR},\mathrm{WI},\mathrm{WD}\right)$$

There are five input neurons and one output neuron. The number of hidden neurons depended on the training effects. W_ij_ represents the link weight between ith input neuron and jth hidden neuron. V_j1_ represents the link weight between jth hidden neuron and the output neuron.

### 3.2. Design of Training Samples

The training cases of the reservoir were used to obtain the training data set shown in Figure 6. The geometry of the reservoir model was 52 km in total length and was divided into two parts: one was a channel linearly spreading from b to B in width and from h to H in height; another was a cuboid of length L, width B and height H. The starting section of the model reservoir was its inlet. The outlet was located at its end section, which was a center circle hole without actual size. 18 simulation cases used as training samples were designed and are listed in Table 1, with various combinations of b, B, L, H and inflow.

The simulation conditions and parameters of the 3DNM mentioned above were applied to the simulation of the designing cases to acquire the training data set. Before training the ISM-RWTS, it was necessary to normalize the data set. The normalization formula was defined as the following:(2)$$\alpha_{i}^{k}=0.9\frac{x_{i}^{k}-x_{i,min}}{x_{i,max}-x_{i,min}}+0.05$$
where $x_{i,max}$ is the maximum value of the ith factor, and $x_{i,min}$ is the minimum value of the ith factor; $x_{i}^{k}$ is the kth non-normalized sample value of the ith factor; and $\alpha_{i}^{k}$ is the kth normalized sample value of the ith factor.

### 3.3. Training of ISM-RWTS

In this study, Newton’s steepest descent optimization technique [31] was used to help train the ISM-RWTS. The algorithm adjusts the weights (W_ij_ and V_j1_ in Figure 4) of each connection in order to minimize the value of the error function by some small amount of the networks [32]. The trained model learns from the data set and repeats this process up to a sufficiently large number of training cycles. In this way, it usually converges to a certain state when the network is trained with given times, and its learning error can reach a given threshold.

The number of the BPNN’s hidden neurons depends on the degree of nonlinearity and the dimensionality of the problem [33]. We varied the number of neurons in the hidden layer from 1 to 20 for each model, and analyzed the computational accuracy of the BPNN under different hidden neurons. The computational accuracy was evaluated through the root mean square error (*RMSE*), which was defined as follows:(3)$$RMSE=\sqrt{\frac{1}{n}\sum_{i=1}^{n}{\left(Z_{i}-{\hat{Z}}_{i}\right)}^{2}}\;$$
where *n* is the number of samples;$\;{\hat{Z}}_{i}$ is the *i*th predictive value and $Z_{i}$ is the *i*th expected value.

The parameter of learning efficiency and training times were set as 0.3 and 10,000, respectively. Take the ISM-RWTS for May as an example. Figure 7a shows the performances of the different numbers of hidden neurons. It can be noticed that when the hidden neuron was 3, the *RMSE* reached its minimum. After that, the intelligent model was verified, and its fitness is shown in Figure 7b. The number of hidden neurons with the smallest *RMSE* in each ISM-RWTS were confirmed as: 3 for May, 5 for June, 3 for July, 6 for August, and 5 for September.

## 4. Evaluation of Model

The performance of a given model and the simulation of its accuracy are evaluated using the following performance measures: maximum absolute error (*MAR*), mean absolute deviation (*MAD*), *RMSE* (Equation (3)), and the coefficient of correlation (*R*), defined as follows:(4)$$MAD=\frac{1}{m}\overset{n}{\underset{i=1}{\sum }}\left|\left(Z_{i}-{\hat{Z}}_{i}\right)\right|\;\;\;$$
(5)$$RMSE_{\%}=\frac{100}{{\bar{Z}}_{i}}\sqrt{\frac{1}{n}\overset{n}{\underset{i=1}{\sum }}{\left(Z_{i}-{\hat{Z}}_{i}\right)}^{2}}$$
(6)$$R=\frac{\sum_{i=1}^{n}\left(Z_{i}-{\bar{Z}}_{i}\right)\left({\hat{Z}}_{i}-{\bar{\hat{Z}}}_{i}\right)}{\sqrt{\sum_{i=1}^{n}{\left(Z_{i}-{\bar{Z}}_{i}\right)}^{2}\sum_{i=1}^{n}{\left({\hat{Z}}_{i}-{\bar{\hat{Z}}}_{i}\right)}^{2}}}$$
where $Z_{i}$ is the *i*th predicted value of water temperature; ${\hat{Z}}_{i}$ is the *i*th observed or measured value of water temperature; ${\bar{Z}}_{i}$ is the average of $Z_{i}$; ${\bar{\hat{Z}}}_{i}$ is the average of ${\hat{Z}}_{i}$; and *n* is the number of the data considered.

The mean absolute deviation, *MAD*, avoids the problem where the errors cancel each other out, so it can accurately reflect the actual prediction error. The *RMSE* describes the average difference between model results and observations in units of the observed value, and it can be normalized (*RMSE*_%_) to provide a relative measure with respect to the mean water temperature. The coefficient of correlation, *R*, provides information on the strength of the linear relationship between the measured and the simulated value. It ranges from 0.0 to 1.0, with higher values indicating better agreement. The combined use of these parameters provide a sufficient assessment of a given model’s performance.

## 5. Verification and Discussion

### 5.1. Verification on Tankeng Reservoir

The characteristic parameters of the Tankeng Reservoir are listed in Table 2. With the ISM-RWTS, we gained the water temperature stratification at 1 km away from the dam as soon as the parameters were inputted into the ISM-RWTS. The water temperature simulated by the ISM-RWTS was the daily mean value, while the measured one was instantaneous and easily fluctuated, because of the diurnal temperature variation in the summer. Consequently, only the water depths below 10 m were considered.

The comparison of the measured results and those simulated by the 3DNM and the ISM-RWTS is summarized in Figure 8. According to the measured results, the water temperature stratification in the reservoir began to appear in May. Then, the temperature difference between the water surface and the bottom kept increasing and reached its maximum in September. For the simulation under the water depth of 10 m, both the 3DNM and the ISM-RWTS showed the process from almost no water temperature stratification to great water temperature stratification in the Tankeng Reservoir, and presented the same trend as the measured data. It can be noticed that the constant-temperature layer was below the water depth of 50 m, with no more than 1 °C of change. The variable-temperature layer was in the water depth range of 10 m to 50 m, in which the water temperature changed by up to 11 °C. The maximum temperature difference was 11.0 °C for the measured value, 9.5 °C for the 3DNM value, and 10.8 °C for the ISM-RWTS value. When comparing the results of the 3DNM and the ISM-RWTS with the measured results, the average errors were 0.74 °C and 0.72 °C, respectively.

The performance indicators of several models are provided in Table 3. All the *R* at 0.915 or greater indicated that these models were acceptable. The performance of the ISM-RWTS over several months was not stable. Analyzing the results of the ISM-RWTS, it performed best in August, with the highest *R* = 0.986 and the lowest *RMSE*_%_ = 2.984%. In July, the ISM-RWTS got relatively bad results. However, the ISM-RWTS in every month (from May to September) clearly outperformed the 3DNM, because they had lower *R* values. These results indicated that the ISM-RWTS was reliable for simulating the water temperature stratification in the reservoir.

In general, both the 3DNM and the ISM-RWTS performed well in the Tankeng Reservoir simulation. However, the ISM-RWTS was much faster and easier than the 3DNM. In this research, with the ISM-RWTS, we only spent about 6 h to get the results in a 4-core, 3.4 GHz computer, including 4 h for simulating designing cases to acquire the training data set and 2 h for training the model, while it took us nearly 300 h to obtain the results of the 3DNM in the same computer, due to its large calculation area and complex meshes.

### 5.2. Discussion

When constructing the topology of the ISM-RWTS, water temperature was chosen as the output so that the temperature at any water depth could be obtained. Sometimes, it is important to evaluate the quantity of low-temperature water resources in the reservoir for the sake of efficiently exploiting low-temperature water resources, so it is necessary to obtain the depth corresponding to the low-temperature water. In this situation, we adjusted the topology of the ISM-RWTS, i.e., the reservoir depth (DR), reservoir area (AR), reservoir capacity (CR), water inflow (WI), and water temperature (WT) were selected as the inputs, and water depth (WD) as the output. Through the same steps mentioned in Section 3, the Intelligent Computation Model of Reservoir Water Temperature Stratification (ICM-RWTS) was established.

The characteristic parameters of the Tankeng Reservoir in Table 2 were input into the ICM-RWTS to calculate the water depths corresponding to 12 °C, 13 °C, 14 °C, 15 °C, 16 °C, 17 °C, and 18 °C, from May to September. By converting the water depth to the local elevation and smoothly connecting the value in each month, the isoline of the water temperature at 1 km in front of the dam was acquired. The performance of the ICM-RWTS compared with the measured results is shown in Figure 9. The water temperature at the bottom remained at 12 °C all the time. The temperature of most of the water in the Tankeng Reservoir was between 12 °C and 18 °C. Compared with the measured data, the elevation values corresponding to 16 °C, 17 °C and 18 °C fit well with the measured results, while the elevation values calculated by the ICM-RWTS corresponding to 13 °C, 14 °C and 15 °C showed relatively big differences from the measured results, especially in September. This was because this part of the water was close to the constant-temperature layer, and a tiny difference in temperature caused huge variations in the elevation.

The evaluation indicators of the ICM-RWTS are also shown in Table 3. Its *MAE* was 9.4 m at 14 °C in September, and the *MAD* = 2.88 m. In comparison with the 3DNM, it can be noticed that the ICM-RWTS had good performance, with *RMSE_%_
*= 4.687% and *R* = 0.966, which means that the ICM-RWTS could calculate the water temperature isoline well.

## 6. Conclusions

The water temperature in reservoirs is influenced by many factors. Considering the climate, the topography of the reservoir, and the inflow and outflow as the most important factors, the Intelligent Simulation Model of Reservoir Water Temperature Stratification (ISM-RWTS) was constructed. When compared with the 3D numerical model (3DNM), the ISM-RWTS clearly outperformed the 3DNM in the application of the Tankeng Reservoir, and generally had lower values of *RMSE_%_* and higher *R* values. The results showed that the ISM-RWTS is a feasible tool that can be used to estimate the water temperature stratification in the reservoir. The advantages of the ISM-RWTS were reflected in its simplicity of use, its well generalized performance, its transient response, and its flexibility for considering different input and output parameters.

The Intelligent Computation Model of Reservoir Water Temperature Stratification (ICM-RWTS) was also discussed, which was obtained by adjusting the input and output values of the ISM-RWTS. By analyzing the calculation results of the ICM-RWTS, it can be concluded that the ICM-RWTS successfully worked out the isoline of the water temperature over months in the reservoir, and had good accuracy (*RMSE_%_
*= 4.687, *R* = 0.966). This fact further confirms that this method is practical and useful to predict the water temperature in reservoirs.

## Figures and Tables

**Figure 1 ijerph-19-13588-f001:**
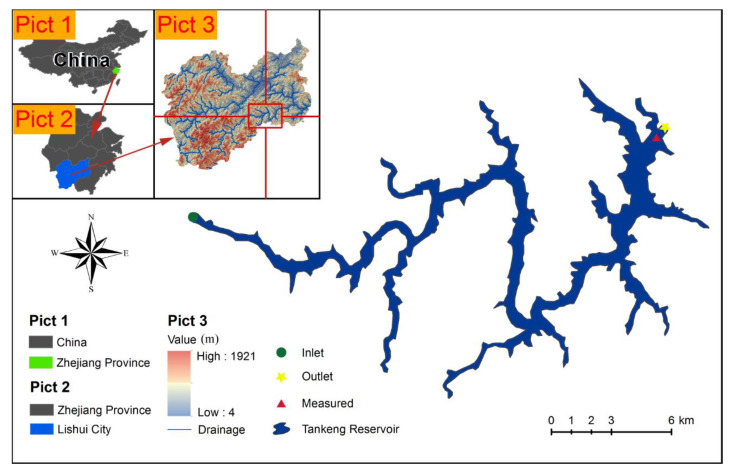
The geographic location of the Tankeng Reservoir. Pict 1, Pict 2 and Pict 3 show the location of the Tankeng Reservoir. The blue represents the water surface at the elevation of 160 m. The yellow star is the basin outlet within the dam. The green dot is the basin inlet 52 km away from the dam. The red triangle indicates the location of the measurement station, 1 km away from the dam.

**Figure 2 ijerph-19-13588-f002:**
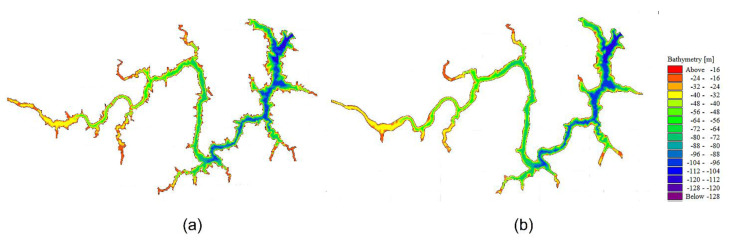
Smoothing on the edge of the calculated region: (**a**) before smoothing, (**b**) after smoothing.

**Figure 3 ijerph-19-13588-f003:**
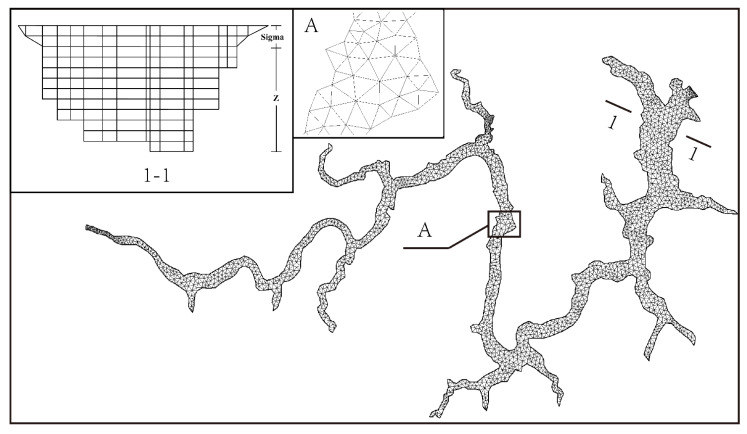
Discretization of 3DNM.

**Figure 4 ijerph-19-13588-f004:**
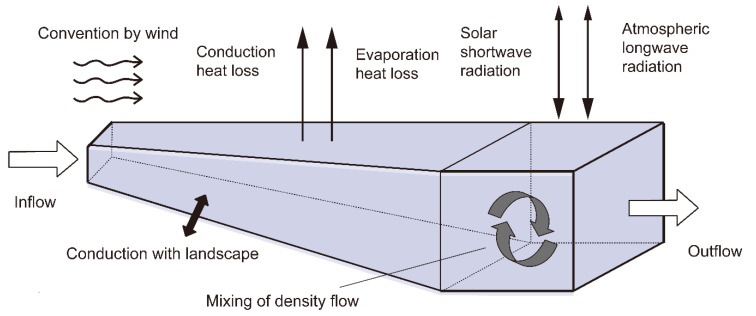
The heat exchange in a reservoir.

**Figure 5 ijerph-19-13588-f005:**
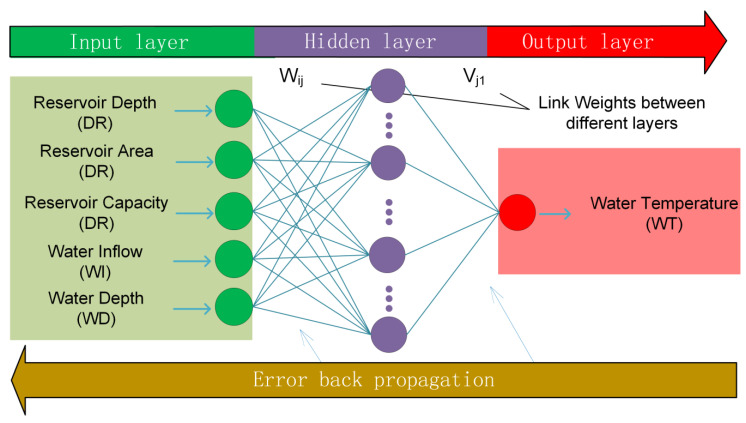
The topological structure of ISM-RWTS.

**Figure 6 ijerph-19-13588-f006:**
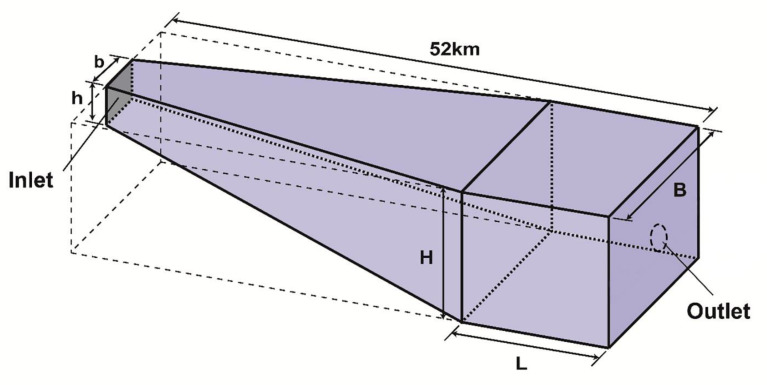
Schematic diagram of reservoir model.

**Figure 7 ijerph-19-13588-f007:**
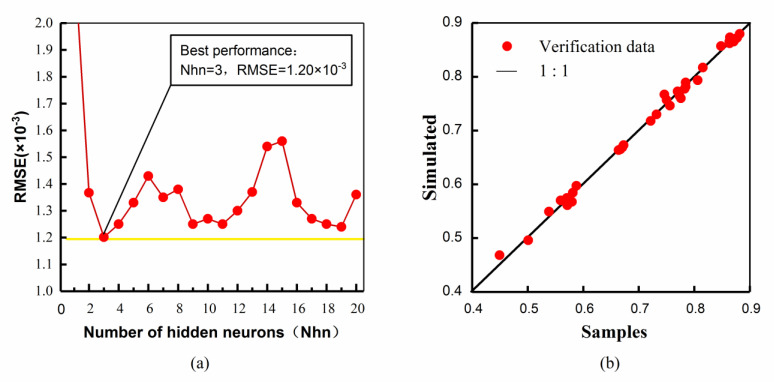
Results of training ISM-RWTS for May: (**a**) the performance of hidden neuron number, (**b**) the test of learning ability.

**Figure 8 ijerph-19-13588-f008:**
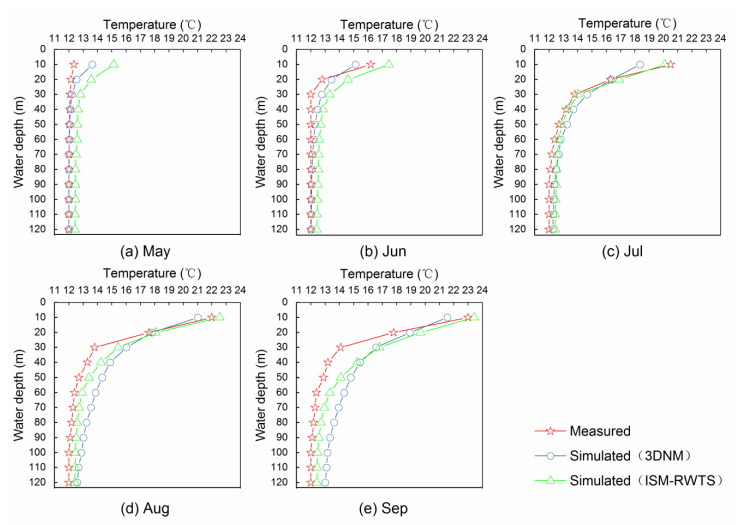
Comparison of measured instantaneous results and simulated by 3DNM and ISM-RWTS: (**a**) May, (**b**) June, (**c**) July, (**d**) August, (**e**) September.

**Figure 9 ijerph-19-13588-f009:**
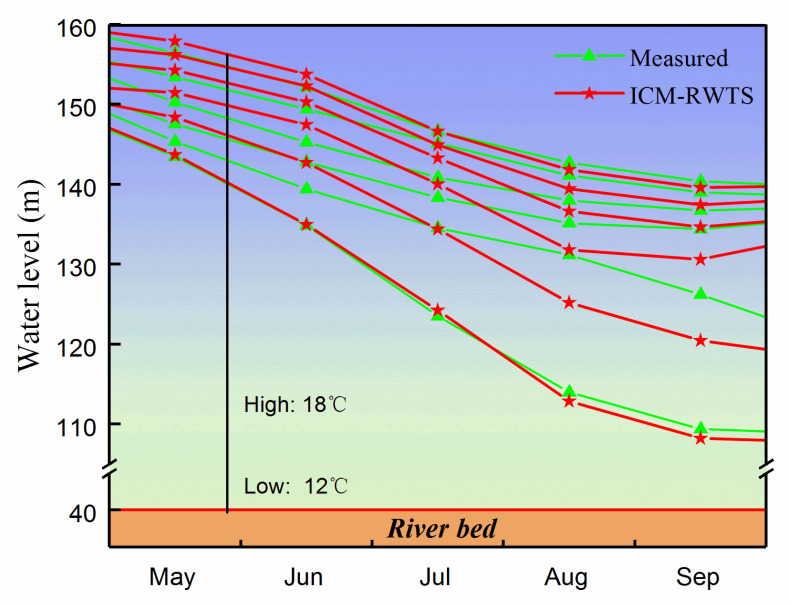
Comparison of water temperature isolines between the measured results and the computed results from ICM-RWTS.

**Table 1 ijerph-19-13588-t001:** Design of training cases.

No.	b (m)	B (m)	L (m)	h (m)	H (m)	Inflow (m^3^/s)	Reservoir Area (km^2^)	Capacity (×10^9^ m^3^)
1	300	800	0	30	120	120	28.60	2.3295
2	400	700	0	30	120	120	28.60	2.2143
3	500	600	0	30	120	120	28.60	2.0776
4	300	800	13,000	30	120	120	31.85	2.9951
5	400	700	13,000	30	120	120	30.55	2.7527
6	500	600	13,000	30	120	120	29.25	2.4942
7	300	800	26,000	30	120	120	35.10	3.6607
8	400	700	26,000	30	120	120	32.50	3.2912
9	500	600	26,000	30	120	120	29.90	2.9108
10	300	800	39,000	30	120	120	38.35	4.3264
11	400	700	39,000	30	120	120	34.45	3.8296
12	500	600	39,000	30	120	120	30.55	3.3274
13	300	800	26,000	30	100	120	35.10	3.0839
14	300	800	26,000	30	80	120	35.10	2.5074
15	300	800	26,000	30	100	100	35.10	3.0839
16	300	800	26,000	30	80	100	35.10	2.5074
17	300	800	26,000	30	100	80	35.10	3.0839
18	300	800	26,000	30	80	80	35.10	2.5074

**Table 2 ijerph-19-13588-t002:** The characteristic parameters of Tankeng Reservoir.

Name	DR (m)	AR (km2)	CR (×10^9^ m^3^)	WI (m^3^/s)	WD (m)
Tankeng Reservoir	120	70.93	4.19	119	10~120

**Table 3 ijerph-19-13588-t003:** Performance indicators of the various models.

MODEL	*MAR*	*MAD*	*RMSE*	*RMSE_%_*	*R*
3DNM	3.630	0.803	1.110	18.97	0.915
ISM-RWTS (May)	2.790	0.779	1.0212	8.46	0.981
ISM-RWTS (Jun)	1.810	0.790	0.896	7.219	0.964
ISM-RWTS (July)	1.650	0.618	0.710	5.179	0.983
ISM-RWTS (Aug)	0.620	0.378	0.401	2.984	0.986
ISM-RWTS (Sep)	2.710	1.018	1.263	9.132	0.975
ICM-RWTS	9.400	2.505	3.513	4.687	0.966

## Data Availability

Not applicable.

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
