# Peer review of "Intelligent Simulation of Water Temperature Stratification in the Reservoir"

_ijerph, 2022, doi:10.3390/ijerph192013588_

Round 1

Reviewer 1 Report

In this manuscript, the authors constructed a 3D numerical model (3DNM) to provide training samples for ISM-RWTS and be used to test its simulation effect. ISM-RWTS is applied on Tankeng Reservoir located in Zhejiang province of China and performed well with an average error of 0.72℃. The use of English is satisfactory and the article can be followed easily. The title accurately reflects the study. The objective is well defined and I have no criticisms regarding the interpretation of results. However, I think that the article is not ready for publication as it stands. The questions are as follows:

1. In the introduction, you need to connect the state of the art to your paper goals. Currently, this is not performed in a convincing way. Please follow the literature review by a clear and concise state of the art analysis. This should clearly show the knowledge gaps identified and link them to your paper goals. Please reason both the novelty and the relevance of your paper goals.

2. To make the conclusion section more clear, authors are highly encouraged to include the point-by-point findings of this article. The current conclusion is written very wide and it is not easy to maintain the key findings.

3. Please review references format, some other refs. can be removed.

Reviewer 2 Report

The paper provides a good explanation of the models being compared with a high level of theoretical detail. The reduction in computational time for the ISM-RWTS compared with the 3DNM is impressive.

The quality of the presentation could be improved if some of the well-established theory could be referenced rather than repeated in the paper. The paper seems rather long.

The study focuses on one location 1 km away from the dam. Please comment on how the model could be applied to model temperature profiles at other locations within the lake or along one of the branches of the lake.

Also, please comment on how applicable the model could be to analyzing reservoirs of a different or asymmetric geometry. Readers may not be familiar with ANN and immediately recognize the effort required to develop a similar model for a different water reservoir.
